# Eigenstate Thermalization Hypothesis
# A Short Review

**Mohsen Alishahiha and Mohammad Javad Vasli**

School of Quantum Physics and Matter
Institute for Research in Fundamental Sciences (IPM),
P.O. Box 19395-5531, Tehran, Iran
alishah@ipm.ir , vasli@ipm.ir

## Abstract

Understanding how an isolated quantum system evolves toward a thermal state from an initial state far from equilibriumsuch as one prepared by a global quantum quench-has attracted significant interest in recent years. This phenomenon can be elucidated through the Eigenstate Thermalization Hypothesis (ETH), which has had a profound impact across various fields, from high-energy physics to condensed matter physics. The purpose of this review article is to present the fundamental concepts of quantum equilibrium and the ETH to a broad audience within the physics community, particularly for those in high-energy physics who seek a comprehensive understanding of these important topics.

*Based on the review talk presented at the "Workshop on Dynamics and Scrambling of Quantum Information," December 2024, IPM.*

## Contents

<sup></sup>12

<sup></sup>13

## 1   Introduction

Based on our everyday experience, the thermalization of macroscopic systems is one of the most natural phenomena in nature. Interestingly, thermalization can also occur in isolated quantum systems. Quantum thermalization is a fascinating concept that has been extensively studied since the early years of quantum mechanics' foundation [1]. Despite significant efforts to explore quantum thermalization, it is still not fully understood.

This is primarily due to the fact that the time evolution of closed quantum systems is unitary and preserves time-reversal symmetry. In principle, one could reverse the unitary evolution to recover the initial state from the final state. In particular, if the final state is a thermal state, one should be able to extract information about the initial state from the thermal state. This creates a contradiction with our understanding of thermalization, as statistical mechanics suggests that equilibrium states do not depend on the microscopic details of the initial states.

One might thus expect that the unitarity of quantum mechanics prevents the relaxation of an initial quantum state into an equilibrated state. This implies that, within the framework of quantum mechanics, true thermal equilibrium should never be reached. Nonetheless, thermalization is consistently observed in everyday experiments, even in systems such as cold atoms.

As we will elaborate later, this challenge can be reframed in terms of quantum chaos, a concept that remains somewhat ambiguous and difficult to understand. The difficulty largely arises from the lack of a clear understanding of quantum phase space. Furthermore, the time evolution of quantum systems is local and unitary, making it difficult to study the emergence of ergodic behavior in quantum systems.

Since thermalization is closely related to quantum chaos, it is useful to discuss chaos briefly. At the classical level, the thermalization of systems is a consequence of the ergodic properties of chaotic classical systems, which justify the use of statistical mechanics in terms of ensembles. A crucial point here is that, for these systems, the ensemble averages used in statistical mechanics calculations match the time averages observed in experiments.

The classical chaotic behavior may be described by the sensitivity of trajectories in the phase space to the initial conditions. For chaotic systems two initially nearby trajectories separate exponentially fast characterized by the Lyapunov exponent. More explicitly, denoting the collective coordinates of the phase space by $Y$, for a chaotic system a small change in the initial time, $Y(0) \to Y(0) + \delta Y(0)$ results in a change in the trajectory of the system at any further time as $Y(t) \to Y(t) + \delta Y(t)$, such that

$$\left( \frac{\partial Y(t)}{\partial Y(0)} \right)^2 \sim e^{\lambda t}, \tag{1}$$

where $\lambda$ is the Lyapunov exponent. This may be compared with that of non-chaotic systems in which the change remains bounded or at most has polynomial growth in time [2]. Note that it order to avoid sign ambiguity we have considered the squared in the above equation.

To go from classical chaos to quantum chaos, one may use the standard quantization procedure. Indeed, taking into account that the left hand side of the equation (1) can be expressed in terms of the Poisson bracket, $\frac{\partial Y(t)}{\partial Y(0)} \sim \{Y(t), Y(0)\}$, the information of quantum chaos may

be encoded in the following quantity

$$\langle ([Y(t), Y(0)])^2 \rangle_\beta \tag{2}$$

that essentially contains the same amount of information as that of the out-of-time-ordered correlator (OTOC)

$$\langle\, Y(t)Y(0)Y(t)Y(0)\,\rangle_\beta\,, \tag{3}$$

which exhibits a non-trivial time behavior. Here $\langle\cdots\rangle_\beta$ denotes thermal average at inverse temperature $\beta$. To be more precise, in a chaotic system consider two operators $O$ and $Q$, then one has

$$\frac{\langle O(0)Q(t)O(0)Q(t)\rangle_\beta}{\langle O(0)O(0)\rangle_\beta \langle Q(t)Q(t)\rangle_\beta} \sim 1 - e^{\lambda_{\mathrm{L}} t}\,, \tag{4}$$

where $\lambda_{\mathrm{L}}$ is quantum Lyapunov exponent which obeys the chaos bound: $\lambda_{\mathrm{L}} \leq \frac{2\pi}{\beta}$ [3].

Although for chaotic systems one expects to see the above time exponential behavior, in general, such a behavior does not necessarily indicate that the system is chaotic. In other words, one may still observe the same behavior even for integrable system which could happen when the potential has a local maximum [4].

The main objective of the present review is to understand thermalization in isolated quantum systems which are generally specified by their Hamiltonian and possibly certain boundary or initial conditions. Therefore, it is natural to expect that the essential information is encoded in the structure of eigenstates and eigenvalues of the Hamiltonian. Indeed, it is known that the eigenvalues of chaotic Hamiltonians exhibit statistical features similar to random matrices. To be precise, *maximally* chaotic Hamiltonians have the same energy level spacing as that of the Random Matrix Theory (RMT) [10] [1].

Let us denote the energy eigenvalues of the Hamiltonian by $E_n$ with the ordering $E_{n+1} > E_n$. The level spacing is defined by $S_n = E_{n+1} - E_n$ and distribution of the level spacing could tell us whether the system is chaotic or integrable. With an *appropriate normalization* of level spacing[2], it is known that if the distribution is Poissonian the model is integrable, whereas for *maximally* chaotic it is Wigner-Dyson distribution. The corresponding distributions are given by

$$\begin{aligned} \text{Poisson} \qquad & P(s) = e^{-s}, \\ \text{Wigner} - \text{Dyson} \quad & P(s) = As^\delta e^{-Bs^2} \quad \text{for } \delta = 1, 2, 4, \end{aligned} \tag{5}$$

for some constant numbers $A, B$ so that $\int P(s) = 1$. Here $P(s)$ is probability density for two neighboring eigenenergies $E_n$ and $E_{n+1}$ having the spacing $s$. Note that for a model with time reversal symmetry one has $\delta = 1$ that is the Wigner surmise $P(s) = \frac{\pi}{2}s e^{-\frac{\pi}{4}s^2}$. For a generic model, the level spacing could lie anywhere between the two extremes of Poisson and Wigner-Dyson distributions. The closer the level spacing is to the Wigner-Dyson distribution, the more chaotic the model is.

To summarize our discussions, intuitively we would expect that equilibrium and thermalization occur for systems that deviate significantly from integrability, both in classical and quantum levels.

In classical mechanics, equilibrium is defined by the infinite time average of an observable,

---

[1]Although RMT is a natural framework to understand quantum chaos and thermalization and even the main motivation of ETH comes from RMT, in this review paper we would rather intentionally not to discuss it. Indeed, in order to make it easier to follow the subject, our main idea is to introduce thermalization and the basic concept of ETH using absolutely minimum background knowledge needed to address this subject. Those who are interested in RMT are referred to the book [8] or the review article [9].

[2]For detail of the definition of "appropriate normalization" and evaluating level spacing see *e.g.* [11, 12] ( see also Appendix A).

though performing the infinite time average is a challenging task. We note, however, that in this case, ergodicity implies that in a sufficiently long period, a chaotic system will explore its entire phase space, allowing the use of the ensemble average as a substitute for the infinite time average.

Although the general idea of equilibrium and thermalization could be extended to the quantum level, in the realm of quantum mechanics, we would expect that the process of thermalization exhibits a distinct feature. Therefore, it is of great interest to study the process of thermalization within quantum mechanics rather in more details.

To proceed, it is worth noting that, at the quantum level, although for closed quantum many body systems one may observe emerging of a thermal equilibrium state in the non-equilibrium dynamics, unlike the classical systems, the thermalization may occur without any time average [13]. Indeed, in this case, the out of equilibrium states approach to their thermal expectations shortly after relaxation. To make the statement more precise, let us consider a closed chaotic quantum system prepared initially in the state $|\psi_0\rangle$ and let it evolves unitarily under a local time reversal symmetric Hamiltonian $H$ to get $|\psi(t)\rangle$ at given time $t$. Then for a local typical observable $O$, one gets

$$\langle\psi(t)|O|\psi(t)\rangle \approx \mathrm{Tr}(\rho_{\mathrm{MC}}O) + \text{small fluctuations} \quad \text{for } t \to \infty, \tag{6}$$

where $\rho_{\mathrm{MC}}$ is density matrix of a microcanonical ensemble defined via the following relation

$$\mathrm{Tr}(\rho_{\mathrm{MC}}H) = \langle\psi_0|H|\psi_0\rangle. \tag{7}$$

Actually, the main question in quantum thermalization is that " how this could possibly happen?" The main purpose of ETH [13,14] is to address this question as we review in this paper.

This paper aims to review the basic concepts of the quantum thermalization in which ETH is in its core. To explore the subject we need first to understand the concept of equilibrium in a closed quantum systems and then conditions by which the equilibrium is thermal.

In this review paper, we will intentionally discuss only those issues which are absolutely necessary to follow the logic of thermalization and will not pay attention to different concepts in this area even though they have provided parts of the motivation and framework for further achievements in this field (such as RMT). Of course, we were careful enough to cover the main points and we would expect that this short review makes the reader ready to read scientific papers in this area. This review requires no special background in the field, except an undergraduate course in quantum mechanics and possibly statistical mechanics. For more details and further discussions in this subject, besides the original papers [13–15] which are always useful to read, the reader is referred to review articles [16–18][3]. We acknowledge that this review article builds upon the work presented in these paper. Their essential insights and frameworks have significantly guided our analysis of ETH and informed our discussion throughout this article. The reader may also find a helpful set of lecture notes on Pappalardi's homepage at [19].

The organization of the paper is as follows. In the next Section, we explore the notion of quantum equilibrium. Then in Section Three, we study the condition under which the quantum equilibrium is thermal. In Section Four we review how ETH can be used to understand thermalization. In Section Six we explore certain features of thermalization which have to do with how fast the thermalization may occur that explore the role of the initial states. The last section is devoted to conclusions. In Appendix A, we explore how to define a properly normalized level spacing. For those readers who want to practice numerical computations, we have presented Mathematica scripts in Appendix B that can be used to produce numerical

---

[3]We acknowledge that there are many interesting research and review papers in this field that we have not included here. We apologize to the authors of those papers for not being able to cite them all.

130  results of the paper. It may be also extended to compute other quantities or to apply for more
131  general Hamiltonian.

## 2  Quantum equilibrium

133  Consider a closed quantum system with a local and time independent Hamiltonian, $H$, whose
134  eigenvalues and eigenstates are denoted by $E_n$ and $|E_n\rangle$ where $n = 1, \cdots, \mathcal{D}$ with $\mathcal{D}$ being the
135  dimension of the Hilbert space of the system.

136      Initially, we prepare the system to be in a (non-equilibrium) state $|\psi_0\rangle$ ($\rho_0$, if it is mixed
137  state), which could be the ground state of a local Hamiltonian. Note that if the state is pure one
138  has $\rho_0 = |\psi_0\rangle\langle\psi_0|$. Since the system is isolated, one may consider the unitary time evolution
139  of the state under the local Hamiltonian $H$ given by Schrödinger equation

$$|\psi(t)\rangle = e^{-iHt}|\psi_0\rangle \tag{8}$$

140  or in terms of density matrix one has

$$\rho(t) = e^{-iHt}\rho_0\, e^{iHt}. \tag{9}$$

141      Generally, we are interested in the late time behavior of the expectation value of a typical
142  observable $O$

$$\langle O(t)\rangle = \langle\psi(t)|O|\psi(t)\rangle, \quad \left(\langle O(t)\rangle = \text{Tr}\left(e^{-iHt}\rho_0\, e^{iHt}\, O\right)\right). \tag{10}$$

143  Being a closed system, the evolution is unitary and therefore this expectation value will never
144  stop evolving for finite systems [4]. Nevertheless, it might be the case that this quantity may
145  oscillate around an equilibrium value with small fluctuations almost all the time after a short
146  relaxation time.

147      The main issue we would like to understand is that in what extent and for what times, a
148  suitable equilibrium ensemble could describe the system. As we have already mentioned, the
149  dynamics of the system is unitary and time reversal invariant and therefore for a finite system,
150  a priori, it is not clear how and in what sense the equilibrium may be reached dynamically.

151      The first concept we should make it clear is "what do we mean by equilibrium in the quan-
152  tum level?" Actually, following the original idea of von Neumann [1] when we are studying
153  equilibrium (thermalization) in isolated quantum systems, we should consider physical ob-
154  servables instead of states or density matrices that describe the whole system. Indeed, it is the
155  expectation value of observables, $\langle O(t)\rangle$, that equilibrates. In the sense that it approaches a
156  constant and remains there for almost most of the time, no matter whether the isolated system
157  is in the pure or mixed state. Of course, for a given quantum system it is not obvious if the
158  equilibrium occurs for the system, at all.

159      To proceed, expanding the initial state in the energy eigenstate $|\psi_0\rangle = \sum_n c_n|E_n\rangle$, the
160  equation (10) reads[5]

$$\langle O(t)\rangle = \sum_{n=1}^{\mathcal{D}}|c_n|^2 O_{nn} + \sum_{n\neq m=1}^{\mathcal{D}} c_n^* c_m O_{nm} e^{i(E_n - E_m)t}, \quad \text{with} \quad \sum_{n=1}^{\mathcal{D}}|c_n|^2 = 1, \tag{11}$$

161  where $O_{nm}$ is the matrix elements of operator $O$ in the energy eigenstates, $O_{nm} = \langle E_n|O|E_m\rangle$.

---

[4]Note that if the initial state is pure, it remains so all the times $\rho(t)^2 = \rho(t)$

[5]Here and in what follows we assume that the spectrum of the Hamiltonian is non-degenerate. Moreover,
$E_n - E_m = E_l - E_k$ is valid for either $(E_n = E_m, E_l = E_k)$ or $(E_n = E_l, E_m = E_k)$.

From this equation, it is evident that the time dependence of the expectation value is reflected in the off-diagonal terms. For a system in which we would expect to see equilibrium, the expectation value should approach a time-independent state, requiring that the off-diagonal terms sum to zero. In other words, equilibrium occurs due to a possible phase cancellation at long times. In this case the equilibrated value is given by the infinite time average of the expectation value of the operator. More precisely, one has[6]

$$\overline{\langle O(t) \rangle} = \lim_{T \to \infty} \frac{1}{T} \int_0^T dt \, \langle O(t) \rangle = \sum_{n=1}^{\mathcal{D}} |c_n|^2 O_{nn} = \text{Tr}(\rho_{\text{DE}} \, O), \tag{12}$$

where $\rho_{DE}$ is the density matrix of the diagonal ensemble

$$\rho_{\text{DE}} = \sum_{n=1}^{\mathcal{D}} |c_n|^2 |E_n\rangle\langle E_n|, \qquad (\text{or } \rho_{\text{DE}} = \sum_{n=1}^{\mathcal{D}} \rho_{nn} |E_n\rangle\langle E_n|). \tag{13}$$

This means that the long time equilibrium state is given by the diagonal density matrix

$$\bar{\rho} = \lim_{T \to \infty} \frac{1}{T} \int_0^T dt \, e^{-iHt} \rho_0 \, e^{iHt} = \rho_{\text{DE}}. \tag{14}$$

Thus, although the system evolves under a unitary dynamic which might be out of equilibrium, for most times, the system looks as it is equilibrated to the diagonal state. Mathematically, the above statement may be rephrased as the fact that for a local observable $O$ after an *initial relaxation* the following quantity is small and remains small for most times ( for a time which grows linearly with the size of the system)

$$|\langle O(t) \rangle - \text{Tr}(\rho_{\text{DE}} \, O)| \ll 1. \tag{15}$$

which is a fluctuation above the equilibrium value. In some cases, the above condition may not be satisfied, nonetheless one could still have a bound on the time average of fluctuations. To be precise, one may find an upper bound on the time average of fluctuation

$$\overline{|\langle O(t) \rangle - \text{Tr}(\rho_{\text{DE}} \, O)|^2}. \tag{16}$$

To find the upper bound we note that from (11) and (12) one finds

$$|\langle O(t) \rangle - \text{Tr}(\rho_{\text{DE}} \, O)|^2 = \sum_{n \neq m=1}^{\mathcal{D}} \sum_{l \neq k=1}^{\mathcal{D}} c_n^* c_m \, c_l c_k^* \, O_{nm} O_{lk}^* \, e^{i(E_n - E_m)t - i(E_l - E_k)t}. \tag{17}$$

which results in (see footnote 5)

$$\overline{|\langle O(t) \rangle - \text{Tr}(\rho_{\text{DE}} \, O)|^2} = \sum_{n \neq m=1}^{\mathcal{D}} |c_n|^2 |c_m|^2 |O_{nm}|^2. \tag{18}$$

Using this expression one can find different bounds on the average of the fluctuations. An immediate bound may be found by noting that

$$\sum_{n \neq m=1}^{\mathcal{D}} |c_n|^2 |c_m|^2 |O_{nm}|^2 \leq \text{Max} |O_{nm}|^2 \sum_{n \neq m=1}^{\mathcal{D}} |c_n|^2 |c_m|^2, \tag{19}$$

---

[6]In our notation the"overline" represents an infinite time average.

182  that, using $\sum_{n=1}^{\mathcal{D}} |c_n|^2 = 1$, results in the following bound

$$\overline{|\langle O(t) \rangle - \text{Tr}(\rho_{\text{DE}} O)|^2} \leq \text{Max}_{n \neq m} |O_{nm}|^2. \tag{20}$$

183  Therefore, we would expect that for a closed quantum system an operator reaches its equilib-
184  rium value, independently of the initial state, if its off-diagonal matrix elements in the energy
185  eigenstates are sufficiently small. We will back to this point in the next section.
186      Another bound for the time average of fluctuation may be found by making used the fact

$$(|c_n|^2 - |c_m|^2)^2 \geq 0 \tag{21}$$

187  so that

$$\overline{|\langle O(t) \rangle - \text{Tr}(\rho_{\text{DE}} O)|^2} \leq \sum_{n \neq m=1}^{\mathcal{D}} \frac{1}{2} \left( |c_n|^4 + |c_m|^4 \right) |O_{nm}|^2 = \sum_{n=1}^{\mathcal{D}} |c_n|^4 (OO^\dagger)_{nn}. \tag{22}$$

188  which, using the fact $(OO^\dagger)_{nn} = \langle E_n|O^2|E_n \rangle \leq |O|^2$,[7] results in the following bound

$$\overline{|\langle O(t) \rangle - \text{Tr}(\rho_{\text{DE}} O)|^2} \leq \sum_{n=1}^{\mathcal{D}} |c_n|^4 |O|^2 = \frac{|O|^2}{\xi}, \tag{24}$$

189  where $|O|^2 = \text{Tr}(O^\dagger O)$ is the norm of the operator and, $\xi$ is inverse participation ratio defined
190  by [20]

$$\xi^{-1} = \sum_{n=1}^{\mathcal{D}} |c_n|^4 = \text{Tr}(\rho_{\text{DE}}^2), \tag{25}$$

191  which is essentially a quantity that measures the number of energy eigenstates contributing
192  to the initial state $|\psi_0\rangle$ and thus, unlike the previous case, this bound depends on the initial
193  state. Note that $1 \leq \xi \leq \mathcal{D}$. In fact, when only one energy eigenstate contributes to the state
194  the inverse participation number is one, while when all energy levels equally contribute to the
195  state it is equal to $\mathcal{D}$. Therefore, for sufficiently large $\xi$ we would expect that the operator
196  reaches its equilibrium value. The inverse participation ratio is also a measure to see how
197  mixed the diagonal density matrix is.
198      To conclude we note that if a closed quantum system relaxes to an equilibrium state the
199  equilibrated value should be given by (12). Actually, this may be interpreted as a generalized
200  Gibbs ensemble for which the integral of motions are provided by the projection operators
201  $P_n = |E_n\rangle\langle E_n|$ that trivially commutes with the Hamiltonian. Therefore, the corresponding
202  Gibbs density matrix is

$$\rho_{\text{G}} = \text{Exp}\left( -\sum_{n=1}^{\mathcal{D}} \zeta_n P_n \right), \tag{26}$$

203  where $\zeta_n = -\ln |c_n|^2$. It is worth recalling that starting with a pure state, it remains pure all
204  the time, even though the operator reaches an equilibrium value.
205      So far we have been trying to explore the notion of equilibrium in a quantum system in
206  which the expectation value of a typical local operator approaches its equilibrium value given

---

[7]For future reference, we note that by employing the completeness of the energy eigenstates, the inequality $\langle E_n|O^2|E_n \rangle \leq |O|^2$ can be expressed as $\sum_m |O_{nm}|^2 \leq |O|^2$. Furthermore, since $|O_{nn}|^2$ is always positive, we can write

$$\sum_{m(\neq n)=1}^{\mathcal{D}} |O_{nm}|^2 \leq |O|^2. \tag{23}$$

by the diagonal ensemble (12) and remains there for most of the time. It is not, however, clear if the equilibrium state is thermal. Actually, equilibration is a generic behavior of a large isolated quantum system which occurs *whenever the off-diagonal matrix elements of observables in energy eigenstates are sufficiently small,* or for a bounded operator *when the initial state is sufficiently delocalized in the energy basis*. Though to get a thermal equilibrium for a closed quantum system one may need further condition on the quantum system. It is the aim of the next section to explore this point.

## 3   Quantum thermal equilibrium

For quantum systems with sufficiently large degrees of freedom, the equilibrium values can be described by a few parameters such as the global temperature, energy and particle number. In this case, we may want to use the notion of *thermal equilibrium*. Actually, inspired by the von Neumann's idea of thermalization, we say *an observable is thermalized* if after some relaxation time its expectation value approaches to a value predicted by a microcanonical ensemble (or canonical ensemble) and remains close to it at most of the time. In other words, *thermalization* is used for an equilibration whose diagonal density matrix, $\rho_{DE}$, in a suitable sense, is indistinguishable from a micro-canonical or canonical density matrix. This is the way the prescription of statistical mechanics appears in quantum systems. Interestingly enough, as far as the thermalization is concerned, it does not matter whether the isolated quantum system is in a pure or mixed state.

To make this statement more precise, let us assume that the system is prepared in an initial state $|\psi_0\rangle$ (or $\rho_0$ in the case it is mixed). Thus the average energy (energy expectation value) of the state is

$$E_0 = \langle\psi_0|H|\psi_0\rangle = \sum_{n=1}^{\mathcal{D}} |c_n|^2 E_n, \quad \left( \text{or } E_0 = \text{Tr}(\rho_0 H) = \sum_{n=1}^{\mathcal{D}} \rho_{nn} E_n \right). \tag{27}$$

To have a description in terms of a microcanonical ensemble the corresponding density matrix of the microcanonical ensemble, $\rho_{\text{MC}}$, to which our system is thermalized, should be defined so that $\text{Tr}(\rho_{\text{MC}}H) = E_0$ which involves averaging over all energy eigenvalues in an energy shell centered at $E_0$ with a width $2\Delta E$. Therefore, the desired microcanonical density matrix may be given by

$$\rho_{\text{MC}} = \frac{1}{d_{\text{MC}}} \sum_{E_\alpha} |E_\alpha\rangle\langle E_\alpha|, \quad \text{with } E_\alpha \in [E_0 - \Delta E, E_0 + \Delta E], \tag{28}$$

where $d_{\text{MC}}$ is the dimension of the shell subspace which is, actually, the number of energy levels in the shell that is assumed to be small compared to the full Hilbert space and still large enough to contain many numbers of microstates. Indeed, the dimension of the microcanonical ensemble is given by the entropy of that system at $E_0$; $d_{MC} \sim e^{S(E_0)}$.

Alternatively, one may consider a description in terms of a canonical ensemble in which the corresponding density matrix is given by

$$\rho_{\text{th}} = \frac{e^{-\beta H}}{Z(\beta)}, \qquad \text{with } Z(\beta) = \text{Tr}(e^{-\beta H}), \tag{29}$$

where $\beta$ is the inverse temperature which can be read from the equation $\text{Tr}(\rho_{\text{th}}H) = E_0$.

Using these ensembles, for a given operator, one has

$$\langle O \rangle_{\mathrm{MC}} = \mathrm{Tr}(\rho_{\mathrm{MC}} O) = \frac{1}{d_{\mathrm{MC}}} \sum_{E_\alpha} O_{nn}, \qquad \langle O \rangle_{\mathrm{th}} = \mathrm{Tr}(\rho_{\mathrm{th}} O) = \frac{1}{Z(\beta)} \sum_{n=1}^{\mathcal{D}} e^{-\beta E_n} O_{nn}, \qquad (30)$$

which may be contrasted with the equation 12 where the expectation value is computed in the diagonal ensemble. Actually, utilizing these definitions a quantum equilibrium is *thermal* if the diagonal ensemble approaches the above microcanonical (canonical) ensemble, so that

$$\mathrm{Tr}(\rho_{\mathrm{DE}} O) = \mathrm{Tr}(\rho_{\mathrm{MC}} O), \qquad (\text{ or } \mathrm{Tr}(\rho_{\mathrm{DE}} O) = \mathrm{Tr}(\rho_{\mathrm{th}} O) ) \qquad (31)$$

which may be interpreted as the *quantum version of ergodicity*, that might be thought of as a consequence of *quantum chaos* [15]. In what follows we would like to understand the thermalization for a closed quantum system and to see in what extend and under which conditions the equation (31) holds.

It is worth noting that if the quantum system is integrable (which may happen when it has local conserved quantities) one should not expect the system to thermalize. Indeed, the constants of motion prevent the system to exhibit full thermalization to the microcanonical ensemble. Nonetheless, one can still expect the system to equilibrate to the diagonal state given these locally conserved quantities that would lead to generalized Gibbs ensemble.

For further use, we note that for systems with an exponentially large density of states due to the canonical factor of $e^{-\beta H}$, the expectation value shows a peak at a specific energy. This behavior closely resembles that of a microcanonical ensemble. To elaborate on this point further, for an arbitrary smooth function $f(E_n)$, we observe that the summation over energy in the canonical ensemble can be replaced by an integral, resulting in the following expression

$$\langle f \rangle_{\mathrm{th}} = \frac{1}{Z(\beta)} \sum_{n=1}^{\mathcal{D}} e^{-\beta E_n} f(E_n) = \frac{1}{Z(\beta)} \int_0^\infty dE \, e^{S(E) - \beta E} f(E). \qquad (32)$$

For large $\mathcal{D}$ and using the fact that entropy is extensive one may evaluate the above integral using saddle point approximation to get $\langle f \rangle_{\mathrm{th}} = f(E_\beta) + \mathcal{O}(\mathcal{D}^{-1})$. Here $E_\beta$ is given implicitly form the equation $S'(E_\beta) = \beta$ with prime denotes derivative with respect to energy. Using the similar procedure one also finds $E_\beta = E_0 + \mathcal{O}(\mathcal{D}^{-1})$.

Let us summarize what we have learned so far. We have seen that in order to reach a thermalization two conditions must be met. The first one is given by the equation (12) which means that the summation of time dependent off-diagonal terms should be zero. The second condition is given by (31) that means that equilibrated value is given by microcanonical (canonical) ensemble.

Looking at these conditions one finds certain apparent conceptual problems. First of all, for generic many-body systems, the energy eigenvalues are typically exponentially close to each other and thus, to make sure that the second sum in (11) approaches zero, one should possibly need to wait for an exponentially long time, though in practice thermalization occurs rather in a relatively short time.

The second issue arises with equation (31), which appears to be contradictory. To further explain this, we observe that equations (12) and (30) show that computing the expectation value of an operator in different ensembles involves evaluating the diagonal matrix elements $O_{nn}$ in the energy basis and summing them with a suitable weight. In the case of the diagonal ensemble, this weight is explicitly determined by the probabilities $|c_n|^2$ of the initial state being in energy eigenstate $|E_n\rangle$, which remain constant over time. However, in other ensembles, the information about the initial state is indirectly encoded rather in a cross-grained way through factors such as energy or temperature of the system. Therefore, it becomes clear that this

equation cannot hold universally. Nevertheless, there are approaches to address these apparent conceptual inconsistencies.

A remedy to solve these problems may be given as follows. We note that matrix elements of the observable $O$ in energy eigenstates, $O_{nm} = \langle E_n|O|E_m \rangle$, in general, depends on energy eigenvalues $E_n$ and $E_m$. It is, however, fair to assume that these matrix elements are smooth functions of $\bar{E}$ and $\omega$ defined as follows

$$\bar{E} = \frac{E_n + E_m}{2}, \qquad \omega = E_n - E_m. \tag{33}$$

For *generic states* which are sufficiently narrow in energy (distribution of $|c_n|^2$ is narrow), one may want to assume that the matrix elements $O_{nm}$ are almost constant and do not vary with the eigenstates (slowly varying function). In other words, at leading order in the size of the energy shell, they are functions of $E_0$. In this case one finds

$$\sum_{E_\alpha \neq E_\beta} |O_{\alpha\beta}|^2 \approx |\tilde{O}_{\alpha\beta}|^2 \sum_{E_\alpha \neq E_\beta} 1 \; = |\tilde{O}_{\alpha\beta}|^2 \, e^{S(\bar{E}+\omega/2)} \quad \text{for } E_{\alpha,\beta} \in [E_0 - \Delta E, E_0 + \Delta E], \tag{34}$$

where $e^{S(\bar{E}+\omega/2)}$ is the dimension of the shell with $S$ being thermal entropy. Using the equation (23), up to a function of $\bar{E}$ and $\omega$, one gets

$$\tilde{O}_{\alpha\beta} \leq |O| \, e^{-S(\bar{E}+\omega/2)/2} \tag{35}$$

resulting in the fact that the off-diagonal matrix elements are exponentially small in the size of the system which makes it evident that exponentially long times may not be needed for relaxation.

Similarly, we make the same assumption for the diagonal elements by which from equation (12) one finds

$$\sum_n |c_n|^2 O_{nn} \approx \tilde{O}_{nn}(E_0) \sum_n |c_n|^2 = \tilde{O}_{nn}(E_0) \tag{36}$$

which makes equation (31) consistent.

To conclude we have seen that closed quantum systems exhibit thermalization if matrix elements of observables in energy eigenstates are smooth slowly varying functions of energy within a narrow energy interval. The next section aims to make this statement more precise.

## 4  Eigenstate Thermalization Hypothesis

So far we have been exploring the concept of thermalization in closed quantum systems. Although we have seen that under certain conditions thermalization could occur, its full understanding is still challenging. Since von Neumann's idea on thermalization, there have been several proposals to explain thermalization. The most famous one is the ETH ansatz which gives an understanding of how an observable thermalizes to its thermal equilibrium value. This ansatz describes how observables behave in many-body systems and how their statistics emerge from the quantum mechanical structure of the system. It gives the essence of thermalization in isolated quantum systems and explains why, under certain conditions, the system can be treated as if it were in thermal equilibrium. According to ETH ansatz for sufficiently complex quantum systems the energy eigenstates are indistinguishable from thermal states with the same average energy.

To formulate ETH, following our notation in the previous section, let us consider closed quantum chaotic systems described by a generic local Hamiltonian $H$ with $\mathcal{D}(\gg 1)$ degrees of freedom. The corresponding energy spectrum (eigenenergy) is denoted by $E_n$ which is as-

sumed to be non-degenerate. Then ETH gives an ansatz for the matrix elements of observables in the energy eigenstates as follows [13, 14]

$$O_{nm} = \langle E_n | O | E_m \rangle = \tilde{O}(E_n)\delta_{nm} + e^{-S(\bar{E})/2} f_O(\bar{E}, \omega) R_{nm}. \tag{37}$$

Here $\bar{E}$ and $\omega$ are the same as those defined in (33) and $S(\bar{E})$ is thermal entropy at energy $\bar{E}$ which is an extensive quantity proportional to the size of the system. It is important to note that $\tilde{O}$ and $f_O$ are smooth functions of their arguments, which ensures that the dependence on energy is well-behaved and allows for the averaging processes that lead to thermal behavior. $R_{mn}$ is a random real or complex variable with zero mean $\overline{R_{nm}} = 0$ and unit variance: $\overline{R_{nm}^2} = 1, \overline{|R_{nm}|^2} = 1$. When the system has time-reversal symmetry, $R_{nm}$ is a real random matrix. For real matrix elements one has $f_O(\bar{E}, -\omega) = f_O(\bar{E}, \omega)$, while for complex matrix elements it is $f_O^*(\bar{E}, -\omega) = f_O(\bar{E}, \omega)$. Indeed, this expression summarizes how the matrix elements of an observable $O$ in energy eigenstates can be decomposed into a thermal part (diagonal term) and a perturbative random part (off-diagonal term).

The properties of $R_{nm}$ as a random variable with zero mean and unit variance are crucial for accurately representing the statistical nature of fluctuations in quantum many-body systems. The presence of $R_{nm}$ introduces randomness into the matrix elements of the observable $O$ between different energy eigenstates. This randomness reflects the inherent fluctuations that occur in a many-body quantum system and is essential for understanding how these fluctuations contribute to thermal behavior. Actually, by ensuring that the random variable $R_{nm}$ captures fluctuations in a controlled manner, ETH can account for how local interactions within a quantum many-body system lead to global equilibrium properties, justifying why isolated systems can exhibit thermal characteristics.

The condition $\overline{R_{nm}} = 0$ ensures that, when averaged over many states, the contributions from these fluctuations do not create a bias in the expectation values of observables. This property is crucial for maintaining the integrity of the thermal description, as it implies that the off-diagonal elements do not systematically distort the average behavior of the system. On the other hand the unit variance condition, $\overline{R_{nm}^2} = 1$ and $\overline{|R_{nm}|^2} = 1$, means that the fluctuations have a standard scale. This normalization allows for a consistent comparison of how fluctuations contribute to the overall behavior of observables across different systems or energies. Essentially, it provides a measure of the strength of the noise introduced by $R_{nm}$.

It is usually assume that $R_{nm}$ follow a Gaussian distribution which provides a natural framework for how quantum states might sample and explore the Hilbert space, leading to thermal averages. To justify this assumption we note that in many-body quantum systems, especially in chaotic regimes, the Hamiltonians can be treated using RMT. Indeed, RMT suggests that the eigenvalues and eigenstates of complex quantum systems exhibit universal properties, including the distribution of matrix elements. For chaotic systems, off-diagonal elements in the energy eigenvalues are typically assumed to follow a Gaussian distribution due to the central limit theorem[8]. It is worth mentioning that numerical studies of various models, such as the quantum Ising model or other interacting particle systems, have shown that the off-diagonal matrix elements exhibit Gaussian-like statistics across a range of parameters, lending support to this assumption.

It is also worth mentioning that if these elements are not Gaussian, the coupling between states can become non-uniform or sparse. This could create scenarios where only a few states dominate the dynamics. Indeed, non-Gaussian off-diagonal elements could indicate a breakdown of quantum ergodicity where certain states become localized and do not interact ef-

---

[8]According to the central limit theorem, regardless of the original distribution of the population from which the samples are drawn, the sampling distribution of the sample mean will tend to follow a normal distribution as the sample size becomes large enough.

361 fectively with others. This localized behavior can lead to many-body localization, where the
362 system fails to thermalize because it cannot explore the necessary phase space to average out
363 fluctuations.

364     The ETH ansatz is sufficient to achieve thermal equilibrium. In other words, assuming
365 ETH one can show that the off-diagonal matrix elements of observable are small and also the
366 resultant thermalized value is consistent with that of the microcanonical ensemble. As we
367 have explored in the previous sections, these are two crucial ingredients required for a system
368 to thermalize. To be precise, by making use of ETH and plugging into the equation (20) one
369 finds

$$\overline{|\langle O(t)\rangle - \text{Tr}(\rho_{\text{DE}}\, O)|^2} = \overline{\langle O(t)\rangle^2} - \overline{\langle O(t)\rangle}^2 \le \text{Max}_{n\neq m}|O_{nm}|^2 \propto e^{-S(\bar{E})}, \tag{38}$$

370 showing that the time fluctuations of the observable's expectation values are exponentially
371 negligible and therefore the expectation value of the observable is very close to its equilibrated
372 value and remains so for most of the time. Thus the first criteria of thermalization is satisfied.

373     Having shown that ETH is enough to reach equilibrium, one can go further to explore how
374 thermal equilibrium arises in this context. To begin, it is useful to compute thermal expectation
375 value $\langle O\rangle_{\text{th}}$ utilizing ETH. Indeed, from (30) one has

$$\langle O\rangle_{\text{th}} = \frac{1}{Z(\beta)} \sum_{n=1}^{\mathcal{D}} e^{-\beta E_n} O_{nn} = \frac{1}{Z(\beta)} \int_0^\infty dE\, e^{S(E)-\beta E} \tilde{O}(E) + \mathcal{O}(e^{-S/2}). \tag{39}$$

376 By making use of the same procedure elaborated in the previous section, one can compute the
377 above integral using saddle point approximation to arrive at

$$\langle O\rangle_{\text{th}} = \tilde{O}(E_0) + \mathcal{O}(\mathcal{D}^{-1}) + \mathcal{O}(e^{-S/2}). \tag{40}$$

378 One could also consider the microcanonical ensemble to derive an expression for the expecta-
379 tion value in this ensemble, given by $\langle O\rangle_{\text{MC}} \approx \tilde{O}(E_0)$. To complete our argument regarding how
380 ETH leads to thermal equilibrium, it remains to be demonstrated that the diagonal ensemble
381 yields the same result within the ETH framework.

382     To proceed, we further assume that $\tilde{O}_{nn}$ is a smooth and slowly varying function of $E$ ( at
383 least over a narrow energy shell supporting the initial state), one may expand it in a Taylor
384 Taylor series about the average energy $E_0$

$$\tilde{O}_{nn}(E_n) = \tilde{O}(E_0) + \sum_{m=1} \frac{(E_n - E_0)^m}{m!}\, \tilde{O}^{(m)}(E_0), \tag{41}$$

385 where $\tilde{O}^{(m)} = \frac{d^m \tilde{O}}{dE^m}$. Therefore, one finds

$$\text{Tr}(\rho_{\text{DE}} O) = \tilde{O}(E_0)\left(1 + \sum_{m=2} \frac{(\Delta E)^m}{m!}\, \frac{\tilde{O}^{(m)}(E_0)}{O(E_0)}\right) + \mathcal{O}(e^{-S/2}). \tag{42}$$

386 where

$$(\Delta E)^m = \sum_{n=1}^{\mathcal{D}} |c_n|^2 (E_n - E_0)^m. \tag{43}$$

387 If the summation in equation (42) is negligible which amounts to assume

$$(\Delta E)^m\, \frac{\tilde{O}^{(m)}(E_0)}{O(E_0)} \ll 1, \quad \text{for } m \ge 2, \tag{44}$$

it can be dropped, leading to our desired result

$$\text{Tr}(\rho_{\text{DE}}O) \approx \tilde{O}(E_0), \tag{45}$$

that fulfills our second condition for thermalization to occur. It is worth noting that in this approximation the matrix representation of a typical operator in energy eigenstates, even diagonal, it is not proportional to the unit matrix. More precisely, one gets

$$O_{nm} \approx \tilde{O}(E_0)\delta_{nm} + \tilde{O}'(E_0)\langle E_n|H - E_0|E_m\rangle + e^{-S(\bar{E})/2}f_O(\bar{E}, \omega)R_{nm}, \tag{46}$$

To conclude, we have seen how ETH is capable to explain thermalization in closed quantum systems that essentially means for an observable $O$ at long times one has

$$\langle\psi(t)|O|\psi(t)\rangle \approx \text{Tr}(\rho_{\text{th}}O) + \text{small fluctuations}, \quad \text{for } t \to \infty, \tag{47}$$

where the small fluctuation is the effect of dephasing. Note that unlike the classical one, here we do not need to use ensemble average. It is very interesting that using ETH, one could show $\overline{\langle O(t)\rangle} \approx \text{Tr}(\rho_{\text{th}}O)$ without making any assumption about the distribution of $|c_n|^2$, except the fact that the variance of energy must be small in the sense expressed in equation (44)[9].

Generally for thermalized systems, one would also expect that there should be thermal fluctuations above the thermal equilibrium value. We note, however, that infinite time average of the fluctuation (20) is too small to account for thermal fluctuations. Nonetheless, one may look at the quantum fluctuations

$$\langle\psi(t)|\left(O - \overline{\langle O(t)\rangle}\right)^2|\psi(t)\rangle, \tag{48}$$

for which one can show

$$\overline{\langle\psi(t)|\left(O - \overline{\langle O(t)\rangle}\right)^2|\psi(t)\rangle} = \overline{\langle O^2(t)\rangle} - \overline{\langle O(t)\rangle}^2 \approx \text{Tr}\left(\rho_{MC}(O - \tilde{O})^2\right) + \mathcal{O}(\Delta E^2), \tag{49}$$

that can be interpreted as the thermal fluctuations. Note that in order to find this relation we have used the fact that matrix elements of $O^2$ in the energy eigenstates have the same structure as that of $O$ given in (37).

It is worth noting that although ETH could provide a framework to understand thermalization, its applicability is limited to few-body or local operators and for states with a finite energy density away from the edges of the spectrum. Therefore, the ground state, low-lying excited states or states with the highest energies for which the energy spectrum becomes less dense are naturally excluded. It does not generally work for integrable systems either.

Essentially the physical content of ETH ( whenever it is applicable) is that, *thermalization in closed quantum systems occurs at the level of individual eigenstates of the Hamiltonian* that means each eigenstate of the Hamiltonian implicitly contains a thermal state. In other words, in quantum systems, the thermal state exists from the beginning and does not emerge by the dynamics of the system. Indeed, the time evolution of the system just unveils the thermal state being hidden by the coherence. The appearance of the thermal state is due to decoherence or dephasing of the off-diagonal terms of matrix elements.

As a final remark, we note that although ETH can provide an explanation or a criterion to determine whether or not a system will eventually thermalize, it cannot explain how it can happen. In fact, how the system evolves into such an "equilibrium" or even thermalizes as a

---

[9]Actually this might require to consider typical initial states which are extensive in energy and sub-extensive in energy fluctuations: $E_0 \sim \mathcal{D}$, $\frac{\Delta E}{E_0} \sim \frac{1}{\mathcal{D}^k}$ for $k > 0$. Alternatively, this might be just a consequence of $\tilde{O}$ being a slowly varying function.

function of time is another important and challenging question.

To get an intuition of how ETH works, it is worth computing rather explicitly the expectation value of local operators in a model that exhibits chaotic behavior. To proceed, let us consider spin$-\frac{1}{2}$ Ising model given by the following Hamiltonian

$$H = -J \sum_{i=1}^{N-1} \sigma_i^z \sigma_{i+1}^z - \sum_{i=1}^{N} (g\,\sigma_i^x + h\,\sigma_i^z). \tag{50}$$

Here $\sigma^{z,y,z}$ are Pauli matrices and $J, g$ and $h$ are constants that define the model. By rescaling one may set $J = 1$, and the nature of the model, being chaotic or integrable, is controlled by constants $g$ and $h$. In particular, for $gh \neq 0$ the model is non-integrable. As we have already maintained by making use of the level spacing distribution one can see whether the model is chaotic or integrable. To see this, setting $g = -1.05$, we have depicted the level spacing for the model (56) for $h = 0.5$ and $h = 0$ in figure 1. For $h = 0.5$ the model is chaotic [21] and, since it is time reversal symmetric, the distribution should follow Wigner surmise, while for $h = 0$ the model is integrable and thus we get Poissonian distribution.

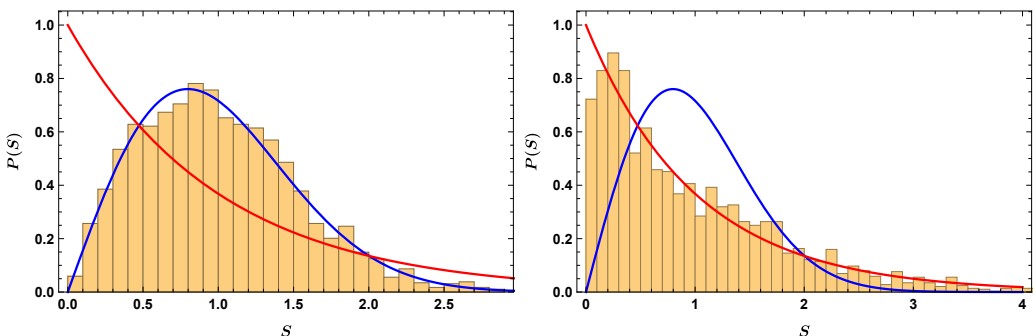

Figure 1: Distribution of the level spacing for the model (56) for $h = 0.5$ (left) and $h = 0$ (right). One observes that for chaotic case it follows Wigner surmise, $P(s) = \frac{\pi}{2} s^{-\pi s^2/4}$, and for integrable case it is Poissonian $P(s) = e^{-s}$.

To illustrate how ETH works, we will compute the matrix elements of $S_x = \sum_{i=1}^{N} \sigma_i^x$ in the energy eigenstates. The results for $N = 8$ are presented in Figure 2. In the non-integrable case, we observe that the diagonal elements are non-zero and significantly larger than the off-diagonal elements, which are predominantly zero. In contrast, this behavior is not observed in the integrable case, as shown in the right panel of Figure 2 (see also Figure 3).

To examine the time dependence of the expectation value of an observable, we consider the magnetization in the $x$-direction, defined as $S_x = \sum_{i=1}^{N} \sigma_i^x$. We compute its expectation value as follows:

$$\langle S_x(t) \rangle = \langle Y + | e^{iHt} S_x e^{-iHt} | Y+ \rangle. \tag{51}$$

The initial state $|Y+\rangle$ is given by

$$|Y+\rangle = \prod_{i=1}^{N} |Y+\rangle_i, \tag{52}$$

where $|Y+\rangle_i$ is the eigenstate of $\sigma^y$ with eigenvalue $+$ at the $i$-th site.

Numerical results for two different scenarios are depicted in Figure 4: one with $h = 0.5$ (a non-integrable model) and the other with $h = 0$ (an integrable model). In the non-integrable case, the expectation value approaches its thermal value and remains close to this value for

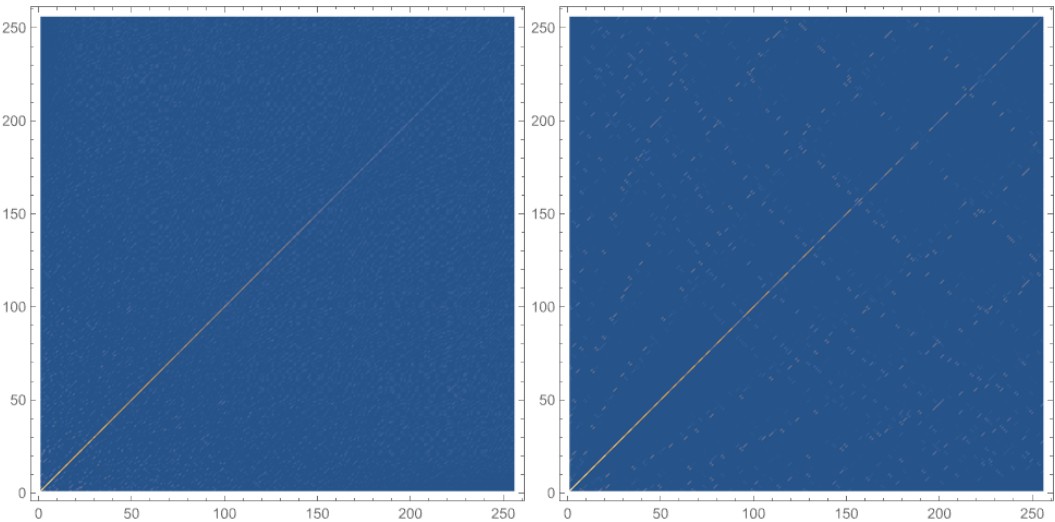

Figure 2: Matrix elements of $S_x$ in energy eigenstates for non-integrable model (left) and integrable model (right). For the integrable case, a closer examination of the figure reveals a clear pattern of non-zero off-diagonal elements. The numerical results are shown for $N = 8$ in the Ising model with $g = -1.05$ and $h = 0.5$ for the non-integrable case and $h = 0$ for the integrable case.

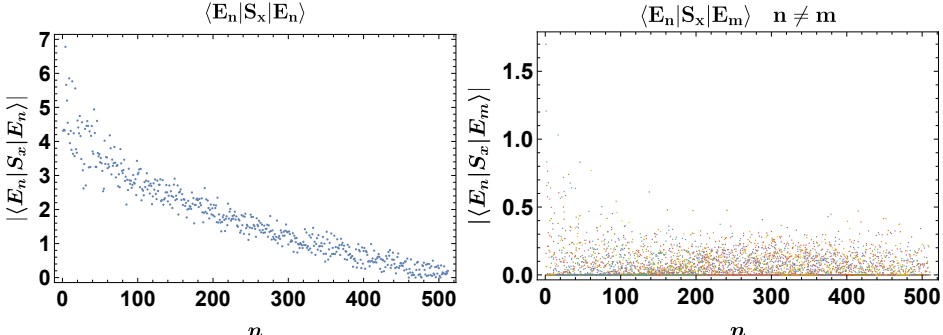

Figure 3: Actual (absolute) values of matrix elements of the operator $S_x = \sum_i \sigma_i^x$ in energy eigenstates for non-integrable case. The numerical results are shown for $N = 9$ in the Ising model with $g = -1.05$ and $h = 0.5$.

an extended period. In contrast, this behavior is not observed in the integrable model, where the expectation value does not stabilize in the same manner.

One might also explore different observables for various initial states. The overall behavior remains consistent with what is illustrated in Figure 4. Thus, we can conclude that for non-integrable systems, after an initial relaxation period, the expectation value of an operator approaches its thermal value and remains there for the majority of the time. As a result, the system becomes indistinguishable from thermal equilibrium. It is important to note that thermalization does not require an ensemble average; it occurs solely through unitary time evolution (see eq. (47)).

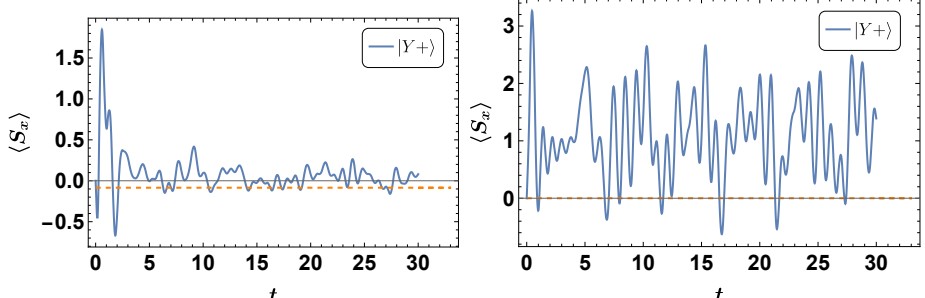

Figure 4: Expectation value of $S_x$ as a function of time for the initial state $|Y+\rangle$ for $h = 0.5$ (left) and $h = 0$ (right). The dashed line is the microcanonical prediction of thermalization. As we see in the non-integrable case the late time value is consistent with the microcanonical predication, confirming the equation (47).

## 5  The role of initial state: Weak and Strong thermalization

Although it is generally believed that a non-integrable model will thermalize, the nature of this thermalization can vary across different scenarios. In addition to the Hamiltonian, which governs the dynamics of the system, the characteristics of thermalization may also depend on the initial state. Consequently, within a fixed model, different initial states can exhibit distinct behaviors [21].

As previously mentioned, our primary objective is to study the time evolution of the expectation value of a local operator (observable) $O$

$$\langle \psi(t)|O|\psi(t)\rangle = \mathrm{Tr}\left(e^{-iHt}\rho_0\,e^{iHt}\,O\right). \tag{53}$$

In the context of chaotic systems, one can examine the nature of thermalization, being strong or weak, by investigating the behavior of the above expectation values. In strong thermalization, the expectation value relaxes to the thermal value rapidly, indicating a quick equilibration process. In contrast, weak thermalization is characterized by significant oscillations around the thermal value, although the time average eventually converges to the thermal value. Understanding these dynamics can provide insights into the mechanisms underlying thermalization, distinguishing between strong and weak thermalization regimes.

Alternatively, one can compute the trace distance between the non-equilibrium state $\rho(t)$ and the thermal state $\rho_{\mathrm{th}}$ using the formula $\mathrm{Tr}(|\rho(t) - \rho_{\mathrm{th}}|)$, which serves as a measure of quantum thermalization. It is expected that this distance will monotonically decrease to zero in the case of strong thermalization. In the weak-thermalization regime, while a decay in the distance can still be observed, it is accompanied by significant fluctuations.

This general behavior has been numerically explored for the model described in equation (50) in [21]. It was proposed that the observation of strong or weak thermalization is closely related to the effective inverse temperature, $\beta$, of the initial state, which can be determined from the following equation:

$$\mathrm{Tr}(\rho_{\mathrm{th}}H) = E_0. \tag{54}$$

Strong thermalization occurs when the effective inverse temperature of the initial states is close to zero, whereas weak thermalization is observed for initial states whose effective inverse temperatures are sufficiently far from zero. Equation (54) also implies that information about the effective inverse temperature can be gleaned from the expectation value of the energy. In

fact, the regime in which strong or weak thermalization may occur can also be identified by the normalized energy of the initial state [23]:

$$\mathcal{E} = \frac{\text{Tr}(\rho_0 H) - E_{\min}}{E_{\max} - E_{\min}}, \tag{55}$$

where $E_{\max}$ and $E_{\min}$ are the maximum and minimum energy eigenvalues of the Hamiltonian. The quasiparticle explanation of weak thermalization suggests that initial states exhibiting weak thermalization reside near the edge of the energy spectrum [24], where ETH may not hold. While the literature primarily considers the normalized energy (55) to study weak and strong thermalization, it is also beneficial to work with the expectation value of energy itself, as it contains the same amount of information as the normalized energy.

To explore this point better let us consider spin$-\frac{1}{2}$ Ising model given by the following Hamiltonian

$$H = \sum_{i=1}^{N-1} \left( J_x \, \sigma_i^x \sigma_{i+1}^x + J_y \, \sigma_i^y \sigma_{i+1}^y + J_z \, \sigma_i^z \sigma_{i+1}^z \right) + \sum_{i=1}^{N} \left( h_x \, \sigma_i^x + h_y \, \sigma_i^y + h_z \, \sigma_i^z \right)$$
$$+ g_x \, \sigma_1^x + g_y \, \sigma_1^y + g_z \, \sigma_1^z. \tag{56}$$

Here, the parameters $(J_i, h_i, g_i)$ are constants that characterize the model's nature, determining whether it is chaotic or integrable. Specifically, when $J_z, h_x, h_z \neq 0$ and all other parameters are set to zero, the system reduces to the model described in (50), which was investigated in [21]. In this case, it was demonstrated that three distinct initial states, where all spins are aligned along the $x, y$, or $z$ directions—denoted as $|X+\rangle, |Y+\rangle$, and $|Z+\rangle$, respectively—exhibit different thermalization behaviors. In particular, it was found that the initial state $|Y+\rangle$ shows strong thermalization. In contrast, the initial state $|Z+\rangle$ displays weak thermalization, while the initial state $|X+\rangle$ appears to deviate significantly from the expected thermal value, suggesting a lack of thermalization for this state. It is worth noting that this apparent deviation in the case of $|X+\rangle$ may be attributed to finite $N$ effects, and even in this scenario, there is evidence of weak thermalization [21, 22].

To further investigate the nature of thermalization in the Ising model described by (56), we consider an arbitrary initial state on the Bloch sphere, which can be parameterized by two angles, $\theta$ and $\phi$, as follows[10]

$$|\theta, \phi\rangle = \prod_{i=1}^{N} \left( \cos \frac{\theta}{2} \, |Z+\rangle_i + e^{i\phi} \sin \frac{\theta}{2} \, |Z-\rangle_i \right), \tag{57}$$

where $|Z\pm\rangle$ are eigenvectors of $\sigma^z$ with eigenvalues $\pm$. Indeed, at each site, the corresponding state is the eigenvector of the operator $\mathcal{O}_i = n \cdot \sigma_i$, with $n$ is the unit vector on the Bloch sphere. More explicitly, one has

$$\mathcal{O}_i(\theta, \phi) = n \cdot \sigma_i = \cos \theta \, \sigma_i^z + \sin \theta \, (\cos \phi \, \sigma_i^x + \sin \phi \, \sigma_i^y), \quad \text{for } i = 1, \cdots, N. \tag{58}$$

For this general initial state and for the model (56) one can compute the expectation value of energy which has the following form

$$E = (N-1) \left( \sin^2 \theta \left( J_x \cos^2 \phi + J_y \sin^2 \phi \right) + J_z \cos^2 \theta \right)$$
$$+ N \left( \sin \theta (h_x \cos \phi + h_y \sin \phi) + h_z \cos \theta \right) + \sin \theta (g_x \cos \phi + g_y \sin \phi) + g_z \cos \theta \tag{59}$$

---

[10] In general the initial state could be identified by $2N$ angles $(\theta_i, \phi_i)$ for $i = 1, \cdots N$. In our case we have assumed that angles in all sites are equal.

Using the analytic expression for the expectation value of energy, we have plotted the energy density, $\frac{E}{N}$, in Figure 5 for $N = 100$ for a model characterized by the parameters $J_z = -1, h_x = 1.05, h_z = -0.5$, with all other parameters set to zero, resulting in a non-integrable system. To emphasize the regions where the energy density approaches zero, we have depicted its absolute value. In this figure, the dark regions indicate initial states that are likely to exhibit strong thermalization. As one moves toward the lighter regions, the degree of thermalization decreases. In particular, the weakest thermalization occurs for the state $|\frac{\pi}{6}, \pi\rangle$. It is worth mentioning that there are several other metrics available to assess weak or strong thermalization (see for example [22–28]).

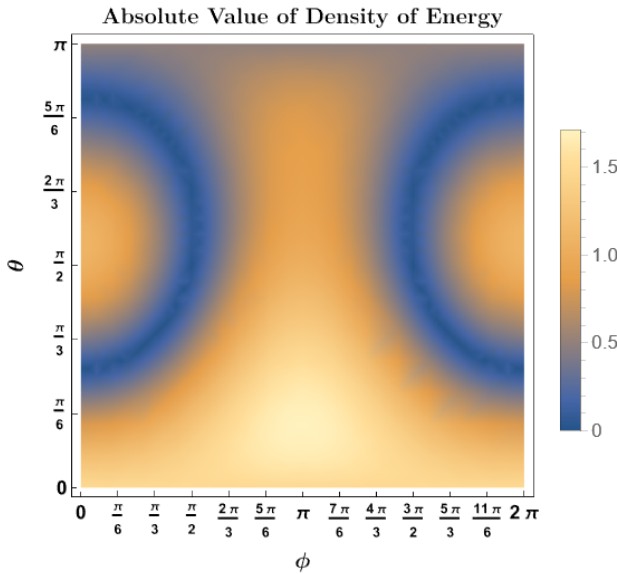

Figure 5: Absolute value of the density of energy evaluated using the analytic expression (59). To draw these plot we have set $N = 100$.

To compare the behavior of the absolute value of the energy density with the effective inverse temperature, we present numerical results for the effective inverse temperature at $N = 9$ in Figure 6. From this figure, it is evident that the behavior of the effective inverse temperature closely matches that of the energy density, despite the fact that the sizes of the two systems used to evaluate these quantities differ by approximately a factor of 15. This demonstrates the robustness of the results with respect to system size. In particular, this finding should be contrasted with results from the literature, where numerical computations have been performed for $N = 14$. Although our effective inverse temperature $\beta$ is evaluated for $N = 9$, the error associated with this comparison is less than 3 percent when compared to the results for $N = 14$.

It is important to note that our understanding of which states exhibit strong or weak thermalization is derived from energy behavior, as shown in the figures 5. From this figure, we observe that the state associated with $\theta = 0$ (arbitrary $\phi$), corresponding to $|Z+\rangle$, exhibits weak thermalization. In contrast, the state at $\theta = \pi$ (arbitrary $\phi$), corresponding to $|Z-\rangle$, shows strong thermalization. As $\theta$ transitions from 0 to $\pi$, a nontrivial behavior emerges, which is evident in the aforementioned figures.

To validate this expectation, we can compute the expectation value of a typical operator across different states to determine if they exhibit strong or weak thermalization. In examining the behavior of these expectation values, we note distinct characteristics for strong and weak thermalization. In strong thermalization, the expectation value rapidly relaxes to the thermal

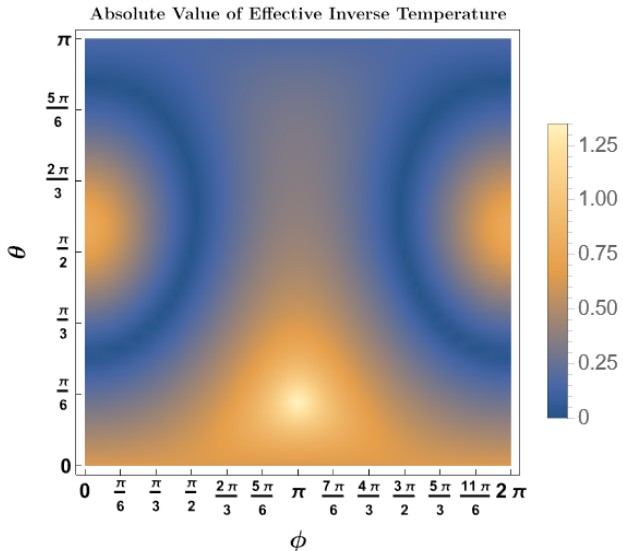

Figure 6: Absolute value of the effective inverse temperature for arbitrary $\theta, \phi$ for the general initial state (57) for Model 1. Here we have set $N = 9$ and $g = 0.5$, $h = -1.05$.

value, while in weak thermalization, it oscillates significantly around the thermal value, even though its time average eventually reaches that value

To get a better understanding of what exactly happens in different points in $\theta - \phi$ plane (initial states), it is useful to explicitly compute the expectation value of local operators to see how thermalization occurs for different initial states. To do so, we will consider the magnetization in the $z$ direction and compute the following quantity

$$\langle S_z(t) \rangle = \langle \theta, \phi | e^{iHt} \sum_{i=1}^{N} \sigma_i^z e^{-iHt} | \theta, \phi \rangle, \tag{60}$$

for different values of $\theta$ and $\phi$. Numerical results reveal two distinct behaviors: in the strong case, we observe a swift relaxation characterized by a quick rise or fall of the expectation value, followed by a saturation phase around the thermal value, with minor fluctuations. Conversely, weak thermalization shows oscillatory behavior from the outset, oscillating around the thermal value.

To quantify these behaviors, we can parameterize the oscillation size (variance of the oscillation) as $\epsilon$ and the amplitude of the first peak or trough following relaxation as $\delta$. The ratio that indicates the strength of thermalization is given by

$$w = \frac{\epsilon}{\delta}. \tag{61}$$

For states exhibiting weak thermalization $w$ is closed to one, while for strong thermalization, $w \ll 1$ (see fig 7). Thus, by assessing the $w$-value for a given initial state, we can determine whether it leans towards strong or weak thermalization. By exploring various initial states, we find perfect agreement with our expectations based on the behavior of energy.

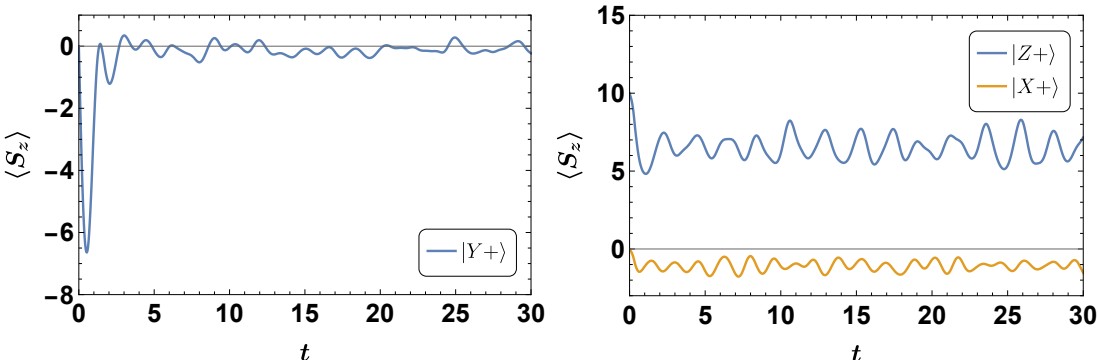

Figure 7: The left panel of this figure shows the expectation value of $S_z$ for the state $|Y+\rangle$, which exhibits strong thermalization, with $w \sim 0.01$. In the right panel, we present the expectation values for the states $|Z+\rangle$ and $|X+\rangle$, both of which demonstrate weak thermalization, with $w \sim 0.5$ and $w \sim 0.7$, respectively.

## 6  Conclusions

In this review paper, we have explored the notion of equilibrium and thermalization for closed quantum systems. To make the notation of equilibrium precise we have followed the original idea of von Neumann in which one looks at physical observables. Actually, what essentially equilibrates is the expectation value of an observable $\langle O(t) \rangle$, in the sense that it approaches a constant and remains there for almost most of the time. Interestingly enough the process does not depend on whether the initial state is pure or mixed.

Although for a given quantum system it is not obvious if the equilibrium occurs, we have seen that if the off-diagonal matrix elements of the observable in energy eigenstates are sufficiently small, it is more likely that the system will relax to an equilibrium state given by the diagonal ensemble. This condition also drops requiring an exponentially long time for relaxation.

We have also seen that thermalization is a special case of equilibrium in which after some relaxation the expectation value of an observable approaches a value predicted by microcanonical ensemble and remains close to it at most of the time. In other words, *thermalization* is used for an equilibration whose diagonal ensemble, is indistinguishable from a microcanonical ensemble whose average energy is the same as that of diagonal ensemble $\mathrm{Tr}(\rho_{\mathrm{DE}} H) = \mathrm{Tr}(\rho_{\mathrm{MC}} H)$.

To make it happen, besides the condition we had imposed on the off-diagonal matrix elements one needs further to impose a condition on the diagonal matrix elements of the observable in energy eigenstates. In fact, we assume that the diagonal matrix elements are almost constant and are independent of particular energy eigenstates. Both of these conditions may be implemented in an ansatz for the matrix elements of local observables known as eigenstate thermalization hypothesis which essentially makes a strong statement that thermalization occurs in the level of each individual energy eigenstate.

Of course, for ETH to work certain conditions must be satisfied. First of all the spectrum of the system must be non-degenerate and the initial state should be typical in the sense it is defined in a range of narrow energy which amounts to have small energy variance in the sense given in (44). Moreover, the operator for which the ETH is given must be local or a sum of a few local operators.

It is worth mentioning that there are several quantities which could be used to probe thermalization and its nature. Although we have not discussed, entanglement entropy is one of the quantities that is usually considered in this context. Let us consider a subsystem denoted by $A$. Then the reduced density matrix is given by $\rho_A(t) = \mathrm{Tr}_{\bar{A}}(\rho(t))$, with $\bar{A}$ being the complement

of the subsystem $A$ and, $\rho(t)$ is the density matrix of the whole system. The system is ergodic if the reduced density matrix of any small subsystem evolves to a thermal density matrix at long times

$$\rho_A(t) = \text{Tr}_{\bar{A}}(\rho(t)) \approx \text{Tr}_{\bar{A}}(\rho_{th}) \qquad \text{for } t \rightarrow \infty. \tag{62}$$

Therefore, the deviation of entanglement entropy $S_A = -\text{Tr}(\rho_A(t) \ln \rho_A(t))$ from volume law at late times is a sign of non-ergodic behavior.

ETH is called strong if all energy eigenstates satisfy ETH ansatz, while it is weak if *almost* all energy eigenstates obey the ansatz. It is then important to take into account the range of the validity of ETH. From the above conditions, it is evident that it should not work where ever the spectrum of energy is sparse, that includes the ground state, low-lying excited states or states with the highest energies.

According to ETH thermalization occurs at the level of energy eigenstates. There are, however, some exceptions for which the system does not exhibit ergodic behavior. This behavior may happen due to certain symmetries resulting in an extensive number of operators that commute with the Hamiltonian which yields many conserved quantities. These conserved charges prevent the system from thermalizing and thus we get strong violation of ETH in the sense that all energy eigenstates violate ETH ansatz. This may happen in integrable models or many body localization [29–31]

Moreover, as we have already mentioned, even for the ergodic cases, ETH may not be valid for certain regimes of the spectrum, that includes near the edges of the spectrum. Then, it is an important question to pose whether there are any other cases where ETH may be violated. Interestingly enough, there are certain states on the bulk of the spectrum which also violate ETH leading to a weakly ETH violation.

In fact a non trivial unexpected observation has been made in an experiment of a chain of Rydberg atoms in [32] where it was observed a significant departure of the expected dynamics depending on the choice of the initial state. Being an ergodic system with no conserved quantities except total energy, one would expect to get behavior predicted by ETH, namely relaxation to the thermal ensemble for general initial states. Nonetheless, it was observed that there are certain states exhibiting periodic revivals (non-thermal states with a robust oscillatory behavior) which can not be explained by integrability and localization as there are no conserved charges and disorder. This behavior has been explored in a particular model known as PXP Hamiltonian [33] and has been interpreted in terms of quantum many-body scars [34, 35] (see also [36, 37]) [11]. Scars are examples of weakly violating ETH which are essentially special eigenstates within the thermalized spectrum, that exhibit non-ergodic behavior.

Quantum many-body scar states have been studied in different quantum many-body models (see citations of [34, 35]. See also the introduction of [39] for a review of the literature). Besides the fact that there are several theoretical studies and experimental works to explore the properties of scar states which could increase our understanding of quantum thermalization, the scar states found useful in different areas of quantum physics such as quantum information theory, see *e.g.* [40–42]. Due to its fundamental importance, it is of great interest to look at generic systems to see whether or not the system has scar states.

We have also examined different quantities which could probe the nature of thermalization whether it is weak or strong. In the strong thermalization, the expectation value relaxes to the thermal value very fast, while for weak thermalization it strongly oscillates around the

---

[11]The name of many-body quantum scar is in analogy with the single-particle scar states [38]. Consider a classical chaotic system for which we would expect that its quantization results in a quantum chaotic system whose eigenstates are expected to be random. Nonetheless, in some cases, there are some eigenstates which exhibit non-thermal ( ergodicity breaking) behavior located around unstable classical periodic orbits. These states are called scars.

thermal value, though its time average attains the thermal value. Although in both cases the system exhibits thermalization, it may happen with different rates. This might be related to the amount of localization of the initial states in energy eigenstates.

# Acknowledgments

We would like to thank Komeil Babaei Velni, Souvik Banerjee, Ali Mollabashi, Mohammad Reza Mohammadi Mozaffar, Reza Pirmoradian, Mohammad Reza Tanhayi, Nilofar Vardian and Hamed Zolfi for discussions on different aspects of quantum chaos and thermalization and comments on the draft. This work is based upon research founded by Iran National Science Foundation (INSF) under project No 4023620.

# A    Normalization of level spacing

As we have already mentioned distributions of level spacing of energy eigenvalues, when *properly normalized*, can be considered as a measure for a quantum system being chaotic or integrable. This appendix aims to define what we mean by *properly normalized*.

As it is clear from the definition of level spacing one should be careful to see whether the Hamiltonian has symmetry. If symmetries exist, the Hamiltonian should be initially block diagonalized based on the conserved charges. This step is required because eigenvalues within distinct symmetry subspaces are uncorrelated. When there are numerous symmetries leading to only a small number of eigenvalues in each subspace, so that the distribution of spectrum becomes meaningless, the system is trivially integrable. Conversely, with a substantial number of energy levels, one can analyze the distribution of spectrum and level spacing. Indeed, this is what we assume.

Let us confine ourselves in a symmetry subspace containing $\mathcal{D}_0 \leq \mathcal{D}$ energy eigenvalues $E_\alpha$, $\alpha = 1, \cdots, \mathcal{D}_0$ with the ordering $E_{\alpha+1} > E_\alpha$. It is evident that the difference between neighboring energy eigenvalues $s_\alpha = E_{\alpha+1} - E_\alpha$ depends on the density of states of the model under study and we would not expect to get universal behavior. Therefore, one needs to normalize the difference by a *local mean density of state* such that the distribution becomes model independent. This procedure, known as spectral unfolding, involves properly rescaling the spectrum.

To explore the idea let us consider an energy interval $\Delta E$ containing $\mathcal{N}$ energy eigenvalues, so that $\Delta E = E_\mathcal{N} - E_1$. Therefore, the local mean density of state for this interval is given by

$$d(E) = \frac{\mathcal{N}}{\Delta E} . \tag{63}$$

The average of level spacing, $\bar{s}$, in this interval is

$$s_{\text{av}} = \frac{\sum_{\alpha=1}^{\mathcal{N}} s_\alpha}{\mathcal{N}} = \frac{\sum_{\alpha=1}^{\mathcal{N}} E_{\alpha+1} - E_\alpha}{\mathcal{N}} = \frac{E_\mathcal{N} - E_1}{\mathcal{N}} = \frac{1}{d(E)} , \tag{64}$$

that clearly is model dependent. To make it model independent one may define unfolded level spacing as follows

$$\hat{s}_\alpha = d(E)s_\alpha, \tag{65}$$

so that the average of level spacing is one. Then, the distribution should be computed for $S_\alpha = \frac{\hat{s}_\alpha}{\hat{s}_{\text{av}}}$ which is believed to exhibit universal features.

It is worth noting that in the above example to explore the procedure of unfolding, for

simplicity, we have assumed that the local mean density of state by which the unfolding is performed, is uniform over the interval. Though, in general one should consider the case where the local mean density of state depends on the eigenvalue for which the difference is computed, $d(E_\alpha)$. Therefore, the problem reduces to define a proper local mean density of state.

Although the procedure for defining the local mean density of states is not unique, the following method introduced in [11] appears to be simple and efficient. Starting with an energy eigenvalue $E_\alpha$, an energy interval centered at $E_\alpha$ is considered, containing $2\Delta$ energy eigenvalues. This results in $\Delta E = E_{\alpha+\Delta} - E_{\alpha-\Delta}$, and the local mean density of states is thus determined as

$$d(E_\alpha) = \frac{2\Delta}{E_{\alpha+\Delta} - E_{\alpha-\Delta}}. \tag{66}$$

One should also provide a prescription for determining $\Delta$. Generally, it could be any number proportional to a fractional power of the total number of energy eigenvalues $\mathcal{D}_0$. Usually, it is set to be the largest integer number smaller than $\sqrt{\mathcal{D}_0}$. Moreover, For this definition to make sense one needs to assume $\alpha - \Delta \geq 1$ and $\alpha + \Delta \leq \mathcal{D}_0$, so that $\alpha = \Delta + 1, \cdots, \mathcal{D}_0 - \Delta$. Interestingly, this procedure automatically removes the edge effects by removing modes near the edges of spectrum[12].

By making use of this local mean density of state the unfolded level spacing and the average of level spacing are given by

$$\hat{s}_\alpha = d(E_\alpha)(E_{\alpha+1} - E_\alpha) = \frac{2\Delta(E_{\alpha+1} - E_\alpha)}{E_{\alpha+\Delta} - E_{\alpha-\Delta}} \qquad \hat{s}_{\mathrm{av}} = \frac{2\Delta}{\mathcal{D}_0 - 2\Delta} \sum_{\alpha=\Delta+1}^{\mathcal{D}_0-\Delta} \frac{E_{\alpha+1} - E_\alpha}{E_{\alpha+\Delta} - E_{\alpha-\Delta}}. \tag{67}$$

Then the normalized level spacing is defined by $S_\alpha = \frac{\hat{s}_\alpha}{\hat{s}_{\mathrm{av}}}$ that exhibits universal behavior. Namely, its distribution follows Wigner surmise for chaotic case and, for integrable it is Poissonian distribution. Note that $2\Delta$ drops in the definition of normalized level spacing.

# B  Mathematica codes

In this supplementary file, we present Mathematica scripts utilized to obtain the numerical results presented in this paper. These scripts may serve as basic illustrations for readers who wish to acquaint themselves with performing calculations in Mathematica. One could also extend these Mathematica scripts for other models or to evaluate other quantities such as inverse participation ratio.

### Hamiltonian and Initial state

To proceed, we should first introduce the Hamiltonian and the initial state used in our numerical computations. To do so, several preliminary definitions are required. First we define $2 \times 2$ identity matrix

```
identity = IdentityMatrix[2];
sigmaAtN[mat_, n_, L_] := Module[{list}, list = Table[identity, {L}];
list[[n]] = mat;
Fold[KroneckerProduct, First[list], Rest[list]]];
```

---

[12]To make sure that we have really removed the edge effect we could consider $\alpha = k\Delta + 1, \cdots, \mathcal{D}_0 - k\Delta$, for $k \geq 1$, so that we are left with $\mathcal{D}\_2k\Delta$ energy eigenvalues.

The second line in the above box shows how to set a Pauli matrix at $n$-th position of a spin chain with length $L$ ( $L$ sites). By making use of this definition, one can introduce $\sigma_n^x, \sigma_n^y$ and $\sigma_n^z$ as follows

```
sigmax[n_, L_] := sigmaAtN[PauliMatrix[1], n, L];
sigmay[n_, L_] := sigmaAtN[PauliMatrix[2], n, L];
sigmaz[n_, L_] := sigmaAtN[PauliMatrix[3], n, L];
```

These are all we need to define a Hamiltonian for a generic Ising model involving $\sigma^{x,y,z}$. In particular, for the model we have considered in this paper

$$H = -J \sum_{i=1}^{N-1} \sigma_i^z \sigma_{i+1}^z - \sum_{i=1}^{N} (g\sigma_i^x + h\sigma_i^z) \tag{68}$$

one write

```
H[L_, g_, h_] := -g Sum[sigmax[i, L], {i, 1, L}] -
h Sum[sigmaz[i, L], {i, 1, L}] -
Sum[sigmaz[i, L] . sigmaz[i + 1, L], {i, 1, L - 1}];
```

Additionally, it is imperative to define the initial state. The initial state is a tensor product of individual single-qubit states at each site. Consequently, we start by defining the single qubit-state and then the tensor product state in the following form

```
singleState[theta_, phi_] :=
Cos[theta/2] {{1}, {0}} + E^(I phi) Sin[theta/2] {{0}, {1}};

tensorProductState[stateVector_, n_] :=
Fold[KroneckerProduct, stateVector, Table[stateVector, {n - 1}]];
```

By making use of these two definitions, one can introduce the general initial state as follows

```
initialState[theta_, phi_, L_] :=
tensorProductState[singleState[theta, phi], L]
```

## Matrix elements of an Operator in energy eigenstates

We should first select an operator for which we would like to compute its matrix elements. The operator we have considered is $S_x$

```
Sx[L_] := Sum[sigmaz[i, L], {i, 1, L}];
```

We need also to find eigenvectors of Hamiltonian

```
EVectorH = Eigenvectors[N[H[L, g, h]]];
```

The precision of this command is the default of Mathematica. If one wants to fix the precision, let's say 10, it should also be added by hand as follows

```
            EVectorH= Eigenvectors[N[H[L, g, h],10]];
```

726

727     The matrix elements of the operator $S_x$ in energy eigenstates can be then computed as
728 follows

```
            OinEVecH =
            ParallelTable[
            ConjugateTranspose[EVectorH[[m]]] . Sx[L] . EVectorH[[n]], {m, 1,
                  Length[EVectorH]}, {n, 1, Length[EVectorH]}];
```

729

730     One can plot the absolute value of matrix elements using the following command ( see
731 Figure 1)

```
            ListDensityPlot[Abs[OinEVecH], PlotRange -> All]
```

732

733 **Expectation Value of an Operator**

734 In Figure 2, we have considered the operator $S_x$

```
            Sx[L_] := Sum[sigmaz[i, L], {i, 1, L}];
```

735

736     To compute the expectation values of the operator, it is necessary to compute eigenvalues
737 and corresponding normalized eigenvectors of the Hamiltonian

```
            EValueH = Eigenvalues[N[H[L, g, h], 10]];
            EVectorH = Eigenvectors[N[H[L, g, h], 10]];
```

738

739     Also, we need to calculate the representation of the operator in the energy eigenstate

```
            OinEVecH =
            ParallelTable[
            N[Dot[ConjugateTranspose[EVectorH[[n]]] . Sx[L] .
            EVectorH[[m]]]], {m, 1, Length[EVectorH]}, {n, 1,
                  Length[EVectorH]}];
```

740

741     Now we have all the ingredients to compute the expectation value of $S_x$ for any specified
742 values of $\theta$ and $\phi$ (initial state) at any time $t$

```
                   Do[Do[listEV[theta, phi] = {};
                   listc = {};
                   For[m = 1, m <= Length[EValueH], m += 1,
                   AppendTo[
                   listc, {m,
                           N[Dot[ConjugateTranspose[EVectorH[[m]]] .
                           Flatten[initialState[theta, phi, L]]]]}]];
                   EVO[t_] :=
                   ParallelSum[
                   E^(I (EValueH[[m]] − EValueH[[n]]) t)*Conjugate[listc[[n, 2]]]*
                   listc[[m, 2]]*OinEVecH[[m, n]], {n, 1, Length[EValueH]}, {m, 1,
                           Length[EValueH]}];
                   For[t = 0, t <= 30, t += 2/10,
                   AppendTo[
                   listEV[theta, phi], {t, EVO[t]}]], {phi, {Pi/2}}], {theta, {Pi/ 2}}];
```

In this code, we have set $\theta = \frac{\pi}{2}$ and $\phi = \frac{\pi}{2}$, consistent with what is depicted in Figure 2. We can plot Figure 2 with this command

```
                   ListLinePlot[{listEV[Pi/2, Pi/2], {{0, 0}, {30, 0}}},
                   FrameTicks −> {{Automatic, None}}, InterpolationOrder −> 1,
                   PlotStyle −> {{Blue}, {Orange, Dashed}}, PlotRange −> All,
                   Frame −> True,
                   FrameLabel −> {"t", "< \!\(\*SubscriptBox[\(S\), \(x\)]\) >"},
                   LabelStyle −> Directive[FontSize −> 10], RotateLabel −> True,
                   FrameStyle −> Directive[Black, Bold]]
```

## Absolute Value of Effective Inverse Temperature

To compute the effective inverse temperature, we need to compute the eigenvalues of the Hamiltonian and a list of eigenvalues that product in a general inverse temperature $\beta$

```
                   EValueH = Eigenvalues[N[H[L, g, h]]];
                   EValueHbeta = −beta*Eigenvalues[N[H[L, g, h]]];
```

Using these lists, we obtain the trace of a canonical ensemble with a general inverse temperature $\beta$

```
                   Trcanonical =
                   ParallelSum[
                   EValueH[[i]] E^EValueHbeta[[i]], {i, 1, Length[EValueH]}]/
                   ParallelSum[E^EValueHbeta[[i]], {i, 1, Length[EValueH]}];
```

Finally, by numerically solving $\text{Tr}(\rho_{th}H) = \text{Tr}(\rho_0 H)$, we can find the absolute value of the effective inverse temperature

```
                   rho[theta_, phi_, L_] :=
                   Dot[initialState[theta, phi, L] .
                   ConjugateTranspose[initialState[theta, phi, L]]];
                   AbsInversetemp =
                   Abs[NSolve[
                   Trcanonical == N[Tr[Dot[H[L, g, h], rho[theta, phi, L]]]],
                   beta, Reals][[1, 1, 2]]]
```

## Level Spacing of Ising Model

In Appendix A, we have demonstrated the method for computing the level spacing for generic models. Here we present the details of the computation of level spacing for the Hamiltonian (50). As previously stated, identifying the model's symmetries is crucial. The Hamiltonian (50) exhibits parity symmetry (see *e.g.* [43]). Within the Ising model framework, parity symmetry signifies that the system's properties are conserved under spatial inversion. For a one-dimensional Ising model, the parity operation, denoted by $\hat{\Pi}$, inverts the qubit sequence, mapping site $i$ to $L + 1 - i$. For this model, reversing the spin order does not alter the system's energy. In other words, reversing a spin sequence does not affect the nearest-neighbor interactions, thereby maintaining the system's symmetry with respect to this parity operation.

To find out the parity of a state, we construct an orthogonal basis for the Hilbert space utilizing the quantum states of spin. These states are typically expressed using basis vectors that correspond to the eigenstates of $\frac{1}{2}\sigma_z$, where the spins oriented upwards in the z-direction are labeled as $|\uparrow\rangle = \begin{pmatrix} 1 \\ 0 \end{pmatrix}$, and those pointing downward are labeled as $|\downarrow\rangle = \begin{pmatrix} 0 \\ 1 \end{pmatrix}$. Using this basis, we can easily investigate the parity of states. . For example the parity pair of the state $|\downarrow\uparrow\downarrow\downarrow\rangle$ is $|\downarrow\downarrow\uparrow\downarrow\rangle$ . In other words, these two states are related to each other by the parity operator, $\hat{\Pi}|\downarrow\uparrow\downarrow\downarrow\rangle = |\downarrow\downarrow\uparrow\downarrow\rangle$ . According to these explanations, the first step is to build the quantum states of the spins

```
eigensystem = Eigensystem[N[H[L, g, h]]];
sortedEigensystem = Transpose[SortBy[Transpose[eigensystem], First]];
SPINQBasis = Permutations[{1, 1, 1, ... , 1, 1, 1, 0, 0, 0, ...,0, 0, 0}, {L}];
```

In the next step, we find which two bases related together with the parity operator

```
For[i = 1, i <= 2^L, i += 1,
pair[i] = Position[SPINQBasis, Reverse[SPINQBasis[[i]]]][[1, 1]]];
```

Using this basis one can determine the parity of any state $|\psi\rangle$. One may expand the state in the following quantum spin basis

$$|\psi\rangle = \sum \mathcal{C}_i |P_i\rangle, \tag{69}$$

with $|P_i\rangle$ being spin basis and $\mathcal{C}_i = \langle P_i|\psi\rangle$. If we assume that the parity of the state $|\psi\rangle$ is even, it is possible to rewrite the expansion as follows

$$|\psi\rangle = \sum \frac{1}{2}\left(\mathcal{C}_i + \mathcal{C}_j\right)|P_i\rangle, \tag{70}$$

which $C_j$ is projection of $\psi$ on the basis $|P_j >$ that is pair to $|P_i >$ with parity operator $C_j = \langle P_i|\Pi|\psi\rangle = \langle P_j|\psi\rangle$. To ascertain the parity of the state, we employ the probability amplitude. Should the probability amplitude equal one, it indicates that our assumption regarding the state's parity is correct, leading us to deduce that the initial state possesses even parity. Conversely, if the probability amplitude differs from one, then the state's parity is deemed odd. This condition can be represented as follows:

```
For[ i = 1, i <= 2^L, i += 1,
If [ 0.8 < Sum[
1/4 ( sortedEigensystem [[2, i, j]] +
sortedEigensystem [[2, i, pair[j]]])^2, {j, 1, 2^L}] < 1.2,
parity[i] = 1, parity[i] = -1]];
```

788

At this point, we are able to categorize the eigenvalues into two groups: those associated
with odd parity eigenvectors and those with even parity eigenvectors

```
listE = {};
listO = {};
For[ i = 1, i <= 2^L, i += 1,
If [ parity [ i ] == 1, AppendTo[ listE , sortedEigensystem [[1, i]]],
AppendTo[ listO , sortedEigensystem [[1, i ]]]]];
```

791

After dividing the eigenvalues into two sectorsnegative and positivebased on parity sym-
metry, we can normalize the eigenvalues within each sector as detailed in Appendix A. For the
positive sector, we normalize the level spacing as follows

```
leveleven = {};
Delta = Floor[ Sqrt[ Length[ listE ]]];
For[ i = 10 Delta , i <= Length[ listE ] - 10 Delta , i += 1,
AppendTo[ leveleven ,
1/( listE [[ i + Delta ]] - listE [[ i - Delta ]]) ( listE [[ i + 1]] -
listE [[ i ]])]];
```

795

```
EigenValueEven = {};
For[m = 1, m <= Length[ leveleven ], m += 1,
AppendTo[ EigenValueEven , ( leveleven [[m]])/Mean[ leveleven ]]];
```

796

We can do the same for the odd parity sector

```
levelodd = {};
Delta = Floor[ Sqrt[ Length[ listE ]]];
For[ i = 10 Delta , i <= Length[ listO ] - 10 Delta , i += 1,
AppendTo[ levelodd ,
1/( listO [[ i + Delta ]] - listO [[ i - Delta ]]) ( listO [[ i + 1]] -
listO [[ i ]])]];
```

798

```
EigenValueOdd = {};
For[m = 1, m <= Length[ levelodd ], m += 1,
AppendTo[ EigenValueOdd , ( levelodd [[m]])/Mean[ levelodd ]]];
```

799

Finally, we can plot the histogram of level spacing

```
Histogram[ EigenValueEven , {0.1}, "ProbabilityDensity", Frame -> True ]
Histogram[ EigenValueOdd , {0.1}, "ProbabilityDensity", Frame -> True ]
```

801

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
