# Peer review of "Eigenstate Thermalization Hypothesis: A Short Review"

_SciPost Physics Lecture Notes_

## Round 1 · Referee Report · Anonymous (Referee 1) · 2025-7-20

Strengths

1- clarity 2-supported by Mathematica codes 3-short with well chosen highlights

Weaknesses

1- very elementary. It might qualify as a text for students 2 - Citations do not point towards additional material that may help the reader to deepen the given subject 3 - traditional, e.g., using level spacings except gap ratio 4 - supported by Mathematica codes - one would expect python/julia instead 5 - limited coverage of the subject 6 - errors - see below

Report

This journal accepts two types of Lecture notes: advanced and elementary. The manuscript provided may qualify as an initial and elementary introduction to the problem. Then the title should be changed to "Eigenstate Thermalization Hypothesis: A Primer" since certainly, the provided material does not qualify as a review on the subject due to a/ just touching the topic b/providing really quite general and elementary coverage of the subject. On the other hand, it is written in a very friendly manner, making it easy to read, and thus could qualify as a primer on the topic.

Unfortunately, several things would need to be corrected. 1. In 9 places, the term "thermal state" is used. It is not defined. It is mentioned in l.22-23 as a source of controversy in the comment on time evolution. This leads to misconceptions like the statement in l.411-417. A thermal state is a mixed state. How one can understand the statement that any initial state "implicitly contains a thermal state"? What does it mean, explicitly? 2. In l.34 we find an incorrect statement that quantum evolution is "local". It is not true. What about long-range interactions? 3. The following discussion (lines 40-63) of quantum chaos is controversial, uses the concept of temperature (beta), which is not defined for Hamiltonian chaos, and is misleading. Several alternative approaches to quantum chaos exist, see works of Peres, books of Haake and Stoeckmann, etc. 4. Discussion of level spacings is not precise, a power coefficient of repulsion delta=1 corresponds to systems with a generalized time reversal invariance (not just time reversal invariance - see Haake book) - such common misunderstandings should not appear in lecture notes, delta=2, 4 are not defined. Here, level unfolding proposed is suboptimal. That is why, since almost 20 years, gap ratios (dimensionless) have been commonly used in many-body context.

I could follow like that but my main objection comes with Section V on weak and strong thermalization. As an example the authors consider a spin chain, claiming that an arbitrary initial state is given by (57). This is not true since (57) assumes the same position on the Bloch sphere for all spins. Secondly, and more importantly, this is a separable state. The foregoing discussion thus considers a small subset of possible initial states. Then Fig.5 and Fig.6 are restricted only to this subset and tell little about the energy density and possible couplings to other states that necessarily affect the thermalization as a separable state becomes entangled during the time evolution.

Additionally, let me point our that Hilbert space shattering seems unrelated to the notion of many-body quantum scars leading to a different ETH breaking mechanism. Both manifestations of non-ergodicity are mixed up in the manuscript.

To summarize, while this primer on ETH is quite easy and accessible to readers, it suffers from a lack of rigor. This is not acceptable for lecture notes.

Requested changes

1 - Define all physical quantities when they appear first 2 - If necessary provide as an annex a table of definitions 3 - Define quantum chaos not on loose discussion of Lyapunov exponents (that are controversial in the quantum realm) but more rigorously and according to common understanding 4 - Remove incorrect statements 5 - Introduce and explain gap ratio statistics on a side with level spacings 6 - Level spacings (or gaps) are not the only necessary characteristics, explain why 7- Besides Mathematica lines - it would be useful to see python or julia codes 8 - Be more precise on the weak and strong thermalization distinction (if you want to include it in the primer) and on quantum many-body scars. 9 - Explain the idea of Hilbert space shattering as yet another mechanism of ETH breaking

Recommendation

Ask for major revision

---

## Round 1 · Referee Report · Anonymous (Referee 2) · 2025-8-15

Strengths

1- pedagogical and mostly clearly written. 2- accessible to general audience. 3- supported by numerical data and Mathematica code. 4- rapid overview of subject

Weaknesses

1- stucture/ordering of some sections could be improved. 2-other notes exist covering this topic in a similar amount of detail. 4-poor referencing in places. 3- many typos/grammatical errors.

Report

The main strength of these notes is that the main sections (Sec. 2-4) are clearly written and should be understandable to readers with minimal knowledge of the field. The concepts of equilibration, thermalisation, and ETH are logically introduced and the important concepts discussed. The arguments are supported by simple numerical plots for energy and time dependence, which is helpful, but could be done a bit more thoroughly. As is stands, these lecture notes provide an overview of ETH at a very basic level, covering the minimum of concepts required to understand the subject.

There are several issues with the current notes. Firstly, there already exist several good introductions to the subject (e.g. Pappalardi's notes, or more comprehensive reviews). There is still a need for a published set of lecture notes; however, what do these notes contribute which other sources don't? The coverage of the basic topics is similar and other sources go into more detail on other topics. The submitted notes are stated to in particular be targeted at "those in high-energy physics who seek a comprehensive understanding...". However, no connection is made to high-energy physics in the text. I do think the current notes provide a clear and accessible introduction to the subject, but it seems like some extra material could be added, perhaps related to connections in high-energy physics, to set them apart from existing introductions.

I think the ordering of several sections can be improved. In particular, I thought the discussion of chaos in the introduction was premature. At this point it is not clear what the connection between chaos and ETH is, and most of the discussion (e.g. OTOCs) is not mentioned again. I think it would be better to properly motivate the ideas of thermalization and equilibration in the introduction, as is done in later sections, and include the discussion of quantum chaos as a small seperate subsection, or even an appendix.

Finally, I found the referencing of the notes extremely poor in several places. For example, in several places statements like " numerical studies of various models ... have shown that the off-diagonal matrix elements exhibit... lending support to this assumption" appear, with no references to the claimed studies. There are many places in the text which would benefit from additional references, and without this the submission does not pass as a review. There are also quite a few typos or grammatical issues in the text, which need to be fixed.

In summary, based on the clarity with which central topics are explained, I think that these notes could be improved to form a clear and accessible introduction to the topic at the lower graduate level; however, some significant changes are required before they pass the SciPost Lecture Notes acceptance criteria.

Requested changes

1- The notes are less a review and more of an introduction at a basic graduate/upper undergraduate level. The title should be "a short introduction", a "first introduction", or similar.

2- In the abstract, high-energy physics is mentioned but never referred to in the text. Either remove this motivation or (better) add some examples in the text which are directly relevant to HEP.

3- In the introduction, the concept of chaos is introduced but it is not clear how it relates to thermalization. The authors say "As we will elaborate later, this challenge can be reframed in terms of quantum chaos", however chaos is barely mentioned in the rest of the text. It is not clear how the concepts discussed in the introduction are relevant to the discussion of ETH. Either make "Chaos" it's own subsection where it can be discussed in more detail, or move the results to the appendix. As is stands I don't see what this discussion really adds.

4- on line 21, the authors state that "unitary time evolution preserves time-reversal symmetry", then on line 78 (and 324) time-reversal symmetry is discussed in the context of level statistics with a completely different meaning. This is confusing and the presise meanings should be clarified.

5- On line 83 the authors state "To summarize our discussions, intuitively we would expect that equilibrium and thermalization occur for systems that deviate significantly from integrability, both in classical and quantum levels." This is not an accurate summary of the previous paragraphs, which merely discuss chaos.

6- On line 56, the thermal average is used without being defined.

  1. In section 2, I found the discussion clear and well motivated. One problem I had was that always writing both the pure state and mixed state versions of equations seems redundant. Just define the mixed state case properly at the beginning (which contains the pure state case) and show the more general version of equations.

  2. In Eq. 30 the sum indices alpha should be n

  3. Around lines 249 intregrability is discussed and related to local conserved quantities -- this is not the definition of intregability. Many systems have locally conserved charges but are non-integrable.

  4. On line 253 the "Generalised Gibbs ensemble" in mentioned, again with no references.

  5. On line 271 it is stated that "in practice, thermalisation occurs on a relatively short timescale". How short? Is this a theoretical or experimental statement? Make precise. Similarly, in 287, a critical assumption used in the rest of the text is that states are "sufficiently narrow in energy". When is this assumption expected to hold? This is usually true experimentally, but the authors should explain this point.

  6. Line 353, a reference is made to numerical studies, without any references provided.

  7. On 361, many-body localization is mentioned, without any references.

  8. Line 413: the line "each eigenstate of the Hamiltonian implicitly contains a thermal state" is too imprecise and should be removed or clarified.

  9. In Fig. 1, what are the system sizes?

  10. In Fig. 2, the off-diagonal structure in the integrable case is actually invisible in printed version of notes. Can the contrast be increased so features are more visible?

  11. I appreciate the authors showing the numerical curves, but I think the data could be analysed more thoroughly. For example, in Fig 3 different system sizes could be shown, and the structure of off-diagonal elements analysed, e.g. how does f(E, omega) behave?

  12. In Section 5 on strong/weak thermalisation I found the discussion was presented in an unhelpful order. Results are presented showing equivalence of energy and temperature, with references to S/W ETH, but the data for S/W ETH is only presented much later. In particular, in 535 it is claimed “non-trivial behaviour emerges which is evident in the aforementioned figures”, which is not true. I think Fig. 7 should be presented early in the section and the energy of states then introduced to explain the results.

  13. On 486, the quasiparticle explanation of weak ETH is mentioned but not explained.

  14. Eq 59 is broken (lies on top of figure number)

  15. 527: "In particular, this finding should be contrasted with results from the literature" - again, no references are provided!

  16. I didn't see the point in using two figures (6&7) to show the similarity between energy and temperature. It seems like a minor point.

  17. In 597, the definition of strong and weak thermalisation appears different to the previous definition.

  18. 637, "This might be related to the amount of localization of the initial states in energy eigenstates." Imprecise, clarify.

  19. There are many grammatical errors and typos throughout the text. The authors should check through the text carefully.

Recommendation

Ask for major revision

  • validity: ok
  • significance: ok
  • originality: low
  • clarity: high
  • formatting: below threshold
  • grammar: below threshold

---

## Editorial Decision

awaiting_resubmission